# First Report on Cardiac Troponin T Detection in Canine Amniotic Fluid [note 1]

**DOI:** 10.3390/vetsci12100952

**Published:** 2025-10-01

**Authors:** Elisa Giussani, Alessandro Pecile, Andrea Pasquale Del Carro, Valerio Bronzo, Silvia Michela Mazzola, Debora Groppetti

**Affiliations:** 1Department of Veterinary Medicine and Animal Sciences, Università degli Studi di Milano, 26900 Lodi, Italy; elisa.giussani@unimi.it (E.G.); alessandro.pecile@unimi.it (A.P.); valerio.bronzo@unimi.it (V.B.); silvia.mazzola@unimi.it (S.M.M.); 2Iunovet-Clinique Vetérinaire Saint Hubert, 06240 Beausoleil, France; info@iunovet.fr

**Keywords:** amniotic fluid, cardiac troponin T, dog, hypoxia, neonate, viability

## Abstract

This study is the first to explore cardiac troponin T in canine amniotic fluid collected at birth. Cardiac troponin T is a regulatory protein involved in heart muscle contraction that is released in response to cardiac injury. In human medicine, it is used as a marker of foetal distress, whereas its role in canine neonatology remains unknown. The primary aim of this research was to determine whether cardiac troponin T is detectable in the amniotic fluid of dogs and, subsequently, to investigate its potential associations with maternal and neonatal factors. Our findings confirmed the presence of cardiac troponin T in canine amniotic fluid and preliminarily suggest possible links with peripartum clinical parameters. Although the clinical relevance of amniotic cardiac troponin T in dogs requires further investigation, this protein may represent a novel non-invasive diagnostic strategy. It could contribute to the early identification of hypoxic conditions in neonates.

## 1. Introduction

Despite the high rate of neonatal mortality and morbidity within the first week of life in dogs, identifying the responsible causes represents a clinical challenge due to the difficulty in reaching a definitive diagnosis, which frustrates both the veterinarian and the breeder or owner [1,2]. Hypoxia, triggered by factors such as dystocia, maternal hypotension, premature placental detachment, umbilical cord torsion, inadequate maternal care, insufficient labor assistance, and improper management of anesthesia during cesarean section, is a common final pathway leading to the death of newborn puppies [3]. Mild hypoxia during labor is a physiological process essential for stimulating the neonate’s autonomous breathing, induced by an increase in carbon oxide (CO_2_) levels [4]. However, this phase must be transient since prolonged or severe hypoxia can lead to serious consequences for neonates, including bradycardia, tissue hypoperfusion, bradypnea, dyspnea, mixed acidosis, hypercapnia, and ischemic cardiac injury until death [1]. A distressed puppy may not be immediately identifiable solely based on clinical evaluations performed at birth. Neonatal veterinary medicine is therefore constantly engaged in the search for practical, rapid, and reliable diagnostic markers that allow for the early identification of fragile newborns, thereby supporting them and improving their chances of survival promptly. Furthermore, in recent years, there has been an increasing interest towards non-invasive sampling methods for assessing neonatal vitality, with a particular focus on substrates other than blood, such as amniotic fluid collected at birth [5,6,7,8].

Cardiac troponin T (cTnT) is a protein that is released into the blood when the heart muscle is damaged [9]. In human medicine, amniotic cTnT has been associated with perinatal hypoxia leading to fetal myocardial ischemia, often resulting from pregnancy complications with neonatal morbidity and mortality [10]. Based on this evidence, we investigated the presence of cTnT in canine amniotic fluid as a potential indicator of myocardial damage secondary to fetal distress. Additionally, this study investigated the possible correlations between cTnT amniotic levels and various maternal and neonatal parameters. As observed in humans, we expected to detect cardiac troponin in the amniotic fluid of dogs, with concentrations negatively correlated with puppy viability.

## 2. Materials and Methods

In dogs, amniotic fluid is considered biological waste, and its collection at birth poses no risk to either the dam or the puppies. However, as part of a broader research project (Linea 2 Groppetti_2016) focused on the diagnostic relevance of canine amniotic fluid, the study was approved by the Ethics Committee of the Università degli Studi di Milano (OPBA_77_2017) before implementation.

### 2.1. Clinical Procedures

In this study, 14 bitches were monitored from mating and throughout pregnancy until parturition, which occurred either naturally or via elective or emergency C-section. Anamnestic and reproductive data, including breed, head morphology (brachycephalic, mesocephalic, dolichocephalic), age, body weight, litter size, and type of delivery, were recorded. Pregnancy monitoring was conducted according to standard procedures, including progesterone measurement, vaginal cytology, ultrasound, and abdominal radiography [11,12]. Before parturition, dogs planned for elective C-section due to ethical and medical reasons (e.g., brachycephalic breed, history of dystocia, abnormal litter size) underwent cardiological analysis and preoperative blood tests, including a complete blood count and biochemical profile.

Both elective and emergency C-sections were performed following the standard anesthetic and surgical protocols, which involve the extraction of newborns no earlier than 15 min after maternal induction with Propofol [11,13,14,15]. During surgery, immediately after uterine incision, with the neonate still inside the intact amniotic sac, amniotic fluid was gently aspirated using a sterile syringe, then placed in labeled tubes and stored at −80 °C until analysis. A volume of 2–10 mL of AF was collected per puppy, depending on breed size.

After birth, each puppy was provided with conventional neonatal care, including suctioning and removal of remaining fetal fluids from the nasal and oral cavities, as well as gentle rubbing to stimulate respiratory effort [16]. An Apgar score between 0 and 14 was assigned within 5 min of birth to quantify neonatal viability by measuring seven parameters, including heart rate. Neonates with a score of ≤4 were classified as critical, those with a score between 5 and 9 as suboptimal, and those with a score ≥ 10 as viable [17]. Puppies requiring neonatal assistance at birth were provided with supplemental oxygen, stimulation of the acupuncture point GV 26, and/or sublingual atipamezole administration. They were categorized within the resuscitated neonatal group and received further intensive care if needed. Each neonate was identified, and its physical characteristics were recorded, including sex, rectal temperature, body weight, malformations and mortality at birth, at 48 h, and at one week of age. In case of C-section, the birth interval, i.e., the time elapsed between the induction of anesthesia and the extraction of each puppy, was also recorded.

### 2.2. cTnT Measurement in Amniotic Fluid

Cardiac troponin T levels in AF were measured using a canine-specific sandwich ELISA kit (LiStarFish, Via Cavour, 35 - 20063 Cernusco Sul Naviglio (MI), Italy), designed for biological fluids. After thawing, AF was centrifuged at 2000–3000 RPM for 20 min, and the resulting supernatant was analyzed according to the manufacturer’s instructions. The test has a sensitivity of 5.19 ng/L, with intra-assay (CV < 8%) and inter-assay (CV < 10%) precision, and a measurement range of 10–2000 ng/L.

### 2.3. Statistical Analysis

Statistical analyses were conducted using the IBM SPSS Statistics program for Windows, Version 29.0 (IBM Corp, Armonk, NY, USA). The concentrations of cTnT in AF were correlated with maternal continuous variables such as maternal age and body weight using Spearman’s rank-order correlation, while maternal categorical variables such as morphotype (brachycephalic; mesocephalic; dolichocephalic), age (≤4 years; >4 years), body weight (≤10 kg; 10–30 kg; >30 kg) and type of parturition (natural; elective C-section; emergency C-section) were analyzed using independent sample *t*-test (for two-group variables) or one-way ANOVA and Kruskal–Wallis test (for comparisons involving more than two groups).

Neonatal parameters such as Apgar score, heart rate, sex, mortality, and birth interval have also been correlated with cTnT concentrations. The Apgar score was analyzed as a categorical variable with one-way ANOVA (≤4; 5–9; ≥10). The independent *t*-test was used to compare amniotic troponin concentrations with heart rate (≤180 bpm; >180 bpm) and mortality at birth, as well as after 2 and 7 days of life, as categorical variables. The non-parametric U-Mann–Whitney test for two independent samples was used to analyze amniotic cTnT concentrations according to neonate sex. The relationship between amniotic cTnT levels and birth interval was analyzed with Spearman’s non-parametric correlation.

Descriptive statistics are presented as mean ± standard deviation (SD). Statistical significance was considered at *p* < 0.05.

## 3. Results

### 3.1. Clinical Outcomes

The fourteen bitches included in this study are detailed in Table 1. All dams were deemed clinically healthy based on diagnostic evaluations.

Forty puppies (21 females and 19 males) were born, with litter sizes ranging from 1 to 16. Nine bitches and sixteen puppies were brachycephalic, while five bitches and twenty-four puppies were mesocephalic. None of the dogs were dolichocephalic. Bitches were 3.3 to 8.5 years old and weighed from 2.4 to 62.7 kg. Elective C-section was performed in 9 bitches, resulting in the delivery of 33 puppies, while emergency C-section was required in 4 bitches, yielding a total of 5 puppies. One dog gave birth to 2 puppies naturally.

All puppies were assessed using the Apgar score system [17], with values ranging from 0 to 14 (10.3 ± 3.5). In particular, three puppies were classified as critical (Apgar ≤ 4), eight as suboptimal (Apgar 5–9), and 29 puppies as viable (Apgar ≥ 10). Heart rate ranged from 0 to 280 bpm (172.2 ± 45.4). Seven puppies required neonatal assistance at birth. Rectal temperature was between 32 and 34.4 °C (32.8 ± 0.8 °C) in 31 puppies, while this data was not collected for the remaining nine puppies. Birth weight ranged from 130 to 545 gr (368.5 ± 113.2). All three Welsh Corgi littermates (ID. 14) died within 48 h of birth due to severe cardiac and/or renal malformations. No other puppies showed neonatal abnormalities at birth. The neonatal mortality rate at birth was 2.5%, that is, one stillborn puppy out of forty. A further seven puppies died within the first 48 h of life due to various causes: the previously mentioned three Welsh Corgi puppies died due to congenital defects incompatible with life; one puppy was accidentally crushed by the dam; a cohabiting adult fatally injured another dog; in the remaining two puppies, the cause of death was undetermined. Thus, the mortality rate of puppies at 48 h from birth was 17.5%. No further deaths occurred between 48 h and 7 days of life.

The time elapsed between anesthesia induction in the dams, and the extraction of puppies (i.e., the birth interval) was calculated for 29 out of 33 puppies born by elective C-section, while it was not recorded for the remaining 4 four puppies. The birth interval ranged from 15 to 46 min (29.3 ± 8.69), depending on litter size.

### 3.2. Amniotic cTnT

Amniotic fluid was collected from all 40 puppies, and cTnT was detected in every sample, with values ranging from 74.1 to 318 ng/L (191.6 ± 66.4).

Canine morphotype significantly affected the amniotic cTnT concentration (*p* = 0.022), with mesocephalic neonates exhibiting higher levels (216.2 ± 62.5 ng/L; *n* = 5 dams and 24 puppies) than brachycephalic ones (154.7 ± 55.3 ng/L; *n* = 9 dams and 16 puppies).

Maternal age and weight were also significantly correlated with amniotic cTnT concentration. Bitches aged ≤ 4 years (5 dams and their 24 puppies) had higher amniotic cTnT concentrations than older ones (9 dams with 16 puppies) that is, 212.3 ± 65.9 ng/L and 160.5 ± 55.6 ng/L, respectively (*p* = 0.012) (Figure 1).

Bitches weighing over 30 kg (2 dams with 18 puppies) had higher amniotic cTnT concentrations than bitches weighing ≤ 10 kg (7 dams with 12 puppies), that is, 222.7 ± 64.3 ng/L and 156.3 ± 55.1 ng/L, respectively (*p* = 0.018) (Figure 2).

A comparison between amniotic cTnT concentration and litter size could not be performed due to the limited variability within our sample. Except for a single litter of 16 puppies, all other litters comprised between 1 and 3 puppies, thereby limiting the statistical power of the analysis.

The type of parturition was statistically correlated with amniotic concentrations of cTnT (*p* = 0.014). For this analysis, data from elective and emergency cesarean sections were compared, while natural parturition was excluded, as only one bitch and her two puppies belonged to this group. The results showed significantly higher cTnT concentrations in elective (198.5 ± 64.4 ng/L; n = 9 dams, 33 puppies) than emergency C-sections (121.7 ± 34.9 ng/L; n = 4 dams, five puppies, Figure 3).

Among the neonatal parameters considered, only the birth interval was positively correlated with cTnT concentration, which increased with the length of the interval (*p* = 0.042, Figure 4).

No significant correlation was found between the Apgar score and amniotic cTnT concentration, although cTnT levels were higher in critical neonates (n = 3 puppies; Apgar score ≤ 4; 229.2 ± 83.9 ng/L) compared to those with higher Apgar scores (n = 37; Apgar ≥ 5; 188.6 ± 51.5 ng/L) (Figure 5).

Similarly, puppies requiring neonatal care (n = 7) tended to have higher amniotic cTnT concentrations than those not undergoing resuscitation (n = 33), 207.9 ± 55.3 ng/L and 188.2 ± 68.8 ng/L, respectively.

Puppies with a heart rate ≤ 180 bpm (n = 21) tended to have higher amniotic cTnT concentrations (209.6 ± 66.6 ng/L) than puppies with a heart rate above this threshold (n = 19; 195.1 ± 61 ng/L).

Females had slightly higher amniotic troponin concentrations than male puppies. Neonates with a rectal temperature below 33 °C at birth tended to have higher amniotic cTnT concentrations (n = 17, 206.9 ± 65.6 ng/L) compared to those with temperatures above this value (n = 14, 198.4 ± 62.9 ng/L). However, none of these differences reached statistical significance.

Lastly, neonatal mortality (unchanged at 48 h and 7 days) was not significantly correlated with amniotic cTnT concentrations that were 194.8 ± 71.5 ng/L in surviving puppies (n = 32) and 179.0 ± 41.4 ng/L in dying ones (n = 8). In the stillborn puppy, amniotic cTnT reached 218.7 ng/L.

## 4. Discussion

Troponins are regulatory proteins classified into three isoforms (C, I, and T), which are involved in the contraction of cardiac and skeletal muscles [18,19]. In particular, cardiac troponin I and T are considered gold standard biomarkers for detecting heart injury in human medicine [19]. Indeed, when the heart muscle is damaged, troponins are released into the bloodstream, enabling early diagnosis. Elevated levels of cTnT in serum and umbilical cord blood of human neonates have also been associated with perinatal distress [18,20,21,22]. Moreover, amniotic cTnT, which does not cross the placental barrier [23], has proven to be a reliable indicator of fetal hypoxia when analyzed by amniocentesis during pregnancy or at C-section in women [10].

Similarly, in veterinary medicine, serum troponin I and T have been reported as useful predictors of cardiac damage in adults and of fetal hypoxia in caprine [24], equine [25], and canine [1] neonates.

This study is the first to report the presence of cardiac troponin T in canine amniotic fluid, detected in all 40 samples collected at birth. Research addressing amniotic cTnT and its correlation with neonatal viability is also very limited in the human literature [10,26]. Some authors have reported that cTnT is undetectable in amniotic fluid from normal pregnancies, but it ranges from 0.01 to 111.6 µg/L in cases of complicated pregnancies [10]. On the contrary, in another study, healthy fetuses showed amniotic cTnT levels above 0.01 µg/L [26]. This discrepancy may be attributable to the different diagnostic methods used, namely conventional assays that detect µg/L concentrations versus high-sensitivity tests capable of measuring ng/L concentrations. To date, there are no other species in which amniotic T troponin has been measured. In our caseload, troponin was detected in the amniotic fluid of all neonatal dogs, with concentrations ranging from 0.07 to 0.3 µg/L (i.e., 74.1–318 ng/L). The lower maximum troponin levels observed in dogs compared to humans may be attributed to species-specific characteristics, such as a greater tolerance to neonatal hypoxia-ischemia in canine newborns. This enhanced adaptability is likely related to earlier cardiopulmonary maturation, which may reflect an intrinsic resistance of canine cardiomyocytes [27,28].

Interestingly, a significant correlation was observed between amniotic cTnT concentration and several clinical variables, including morphotype, maternal age and weight, type of parturition, and the birth interval, highlighting its promising utility in canine neonatal diagnostics.

Surprisingly, brachycephalic puppies had significantly lower amniotic troponin levels than mesencephalic ones (*p* = 0.022). Considering the predisposition of brachycephalic breeds to dystocia along with the genetic defects in placental vascularization and the reduced neonatal viability observed in their neonates [29,30,31], an opposite trend was foreseen, with elevated cTnT concentrations in brachycephalic dogs. However, the uneven distribution of maternal weight within our canine population may have introduced a confounding effect on amniotic cTnT concentrations, which could explain this result.

Similarly, the influence of maternal weight on maternal age, type of delivery, and neonatal mortality should not be ignored. In fact, since cTnT levels were significantly higher in neonates born to large sized-bitches (>30 kg vs. ≤10 kg; *p* = 0.018), and brachycephalic (11.2 ± 10.7 kg) bitches were lighter than mesencephalic ones (49.9 ± 20.2 kg), maternal weight rather than morphotype may have affected these findings. To date, no studies in either humans or animals have investigated serum or amniotic troponin concentrations in relation to maternal body weight. This aspect may be particularly relevant in dogs, given the wide range of sizes and breeds.

Similarly, younger bitches showed unexpectedly higher values of amniotic troponin concentrations than dams older than 4 years (*p* = 0.012). Although the role of maternal age on serum troponin concentrations remains unclear even in human medicine [32,33], it is reasonable to hypothesize that older bitches may have higher amniotic troponin levels than younger ones, due to their predisposition to pathological pregnancies [2]. However, in our caseload, only two bitches were older than 6 years, and they gave birth to a total of two puppies. Moreover, maternal weight was greater in bitches ≤4 years old (43.6 ± 27.7 kg) than in older bitches (20.7 ± 13.7 kg).

Contrary to our initial assumption, amniotic cTnT concentrations were lower in puppies (n = 5) delivered by emergency compared to elective C-sections (n = 33; *p* = 0.014), while natural parturition was excluded due to it only involved one bitch and two puppies. Once again, however, it should be noted that Chihuahuas, i.e., small dams, mainly underwent emergency C-section.

Stillborn puppies and puppies dying within 48 h of life tended to have a higher value of amniotic cTnT than surviving ones. In this case as well, maternal weight may have contributed to the lack of statistical significance, as bitches of surviving puppies were heavier than those of non-surviving ones, 37.6 ± 26.2 kg and 21.9 ± 19.1 kg, respectively.

As previously mentioned, litter size was not analyzable due to the caseload composition, with all litters consisting of 1 to 3 puppies, except for a single litter of 16 Bernese Mountain Dog puppies, unusually large even by breed standards. In the latter, the average amniotic cTnT concentration (232.6 ± 30.2 ng/L) was higher than in the other litters (164.3 ± 56.2 ng/L). While this data cannot be generalized, the excessive number of fetuses may be attributed to detrimental proper placental perfusion, leading to fetal distress due to a reduced supply of nutrients and oxygen [34].

Excluding natural parturitions, the extraction time for the neonates delivered by elective C-sections (measured from induction to delivery) was positively correlated with amniotic cTnT concentrations (*p* = 0.042), supporting the evidence that prolonged in utero exposure may compromise neonatal viability by increasing the risk of hypoxia, as commonly observed in dystocia [4].

No significant differences in troponin concentrations were observed among viable, suboptimal, and critically ill puppies, including those requiring resuscitation. However, from a clinical perspective, it is noteworthy that neonates with an Apgar score ≤ 4 (n = 3), those with a heart rate ≤ 180 bpm (n = 21), puppies with rectal temperature < 33 °C, and puppies in need of assistance at birth (n = 7) had cTnT concentrations exceeding 200 ng/L. Furthermore, it should be considered that even minimal serum increases in cTnT are indicative of myocardial damage, both in humans and dogs [35,36].

Due to its limited sample size and the primary objective in detecting troponin in amniotic fluid in dogs, our study does not allow for the determination of a troponin cutoff value that could reliably differentiate between healthy and pathological neonates. Nevertheless, we expect that a larger dog population could provide this critical diagnostic parameter.

In humans, the physiological range for blood troponin T concentration is twice as high in men compared to women [37,38]. On the contrary, in our study, amniotic cTnT levels were slightly higher in female puppies. However, dire

ct comparisons with adults may be inappropriate, as the hormonal environment at birth is largely similar between male and female neonates.

As previously highlighted, a key limitation of this pilot study is its small sample size, which comprises a cohort of female dogs with varying breeds, morphotypes, body weights, ages, and litter sizes. These parameters are interrelated variables that may confound each other’s effects on amniotic cTnT concentrations. However, such a composition of the caseload (a relatively small and unbalanced sample size) prevented us from performing a multivariate analysis without the risk of overfitting and spurious associations. Consequently, additional research, including a broader range of breeds, intermediate litter sizes, and a greater number of pathological pregnancies, is warranted further to investigate the clinical and diagnostic potential of amniotic cTnT. While this constraint reduces the statistical power of the findings and should be considered in their interpretation, it does not diminish the potential relevance of amniotic cTnT as a valuable tool for early diagnosis of hypoxia, anemia, and other perinatal disorders in newborn puppies.

A final consideration concerns the molecular weight of cTnT, which is nearly identical in humans and dogs (approximately 37 kDa) [39], suggesting a similar inability of this protein to cross the placental barrier. This aspect supports a fetal, rather than maternal, origin of amniotic cTnT, confirming its specificity and diagnostic value as a neonatal biomarker.

## 5. Conclusions

This pilot study aimed to evaluate the presence of cardiac troponin T in canine amniotic fluid, as no data currently exist on this species. In agreement with human evidence, high concentrations of amniotic cTnT may be associated with hypovital puppies. However, further research is necessary to clarify the diagnostic relevance of cTnT in various clinical contexts. A larger sample size is essential to address the discrepancies observed in comparison to human literature and expected outcomes. Despite its limitations, these preliminary results may offer a novel contribution to assessing neonatal well-being in dogs. While cTnT levels alone cannot identify the precise cause of myocardial injury or fetal hypoxia, defining a specific cut-off value for amniotic cTnT could support the early identification of critical neonates and enable prompt intervention. In future studies, validation of a threshold supported by sensitivity and specificity analyses could facilitate the implementation of preventive measures, potentially reducing postpartum morbidity and mortality in puppies. Thus, although preliminary, our findings open the way to further investigation into the diagnostic potential of amniotic cTnT as a biomarker of fetal distress.

## Figures and Tables

**Figure 1 vetsci-12-00952-f001:**
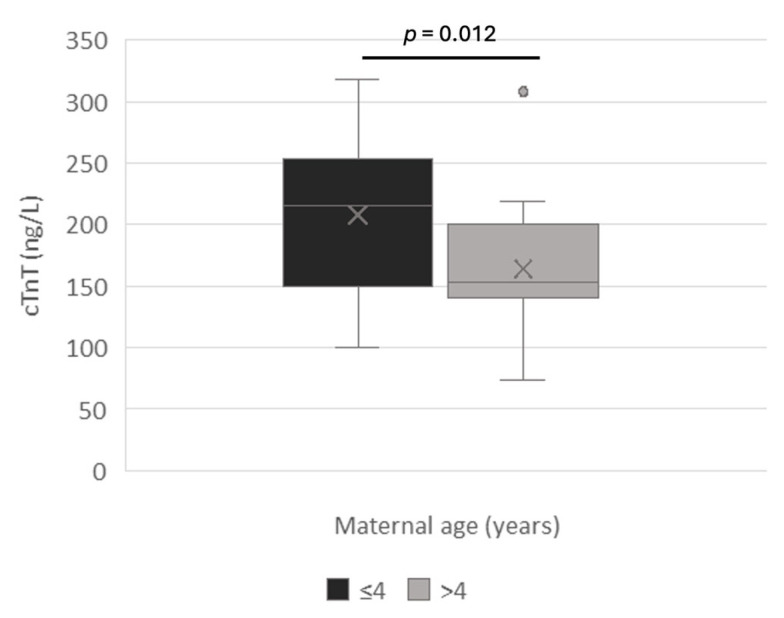
cTnT amniotic concentrations in relation to maternal age.

**Figure 2 vetsci-12-00952-f002:**
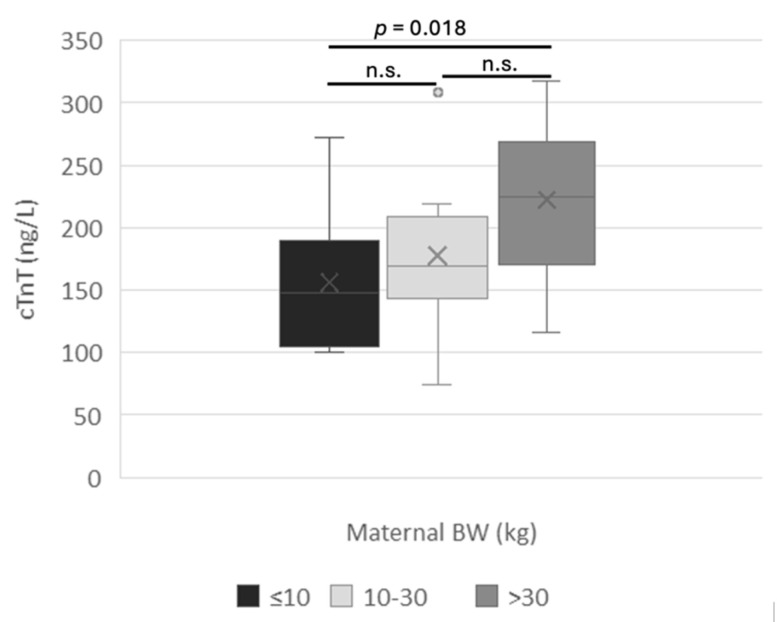
cTnT amniotic concentrations in relation to maternal body weight. BW means body weight: n.s. means not significant.

**Figure 3 vetsci-12-00952-f003:**
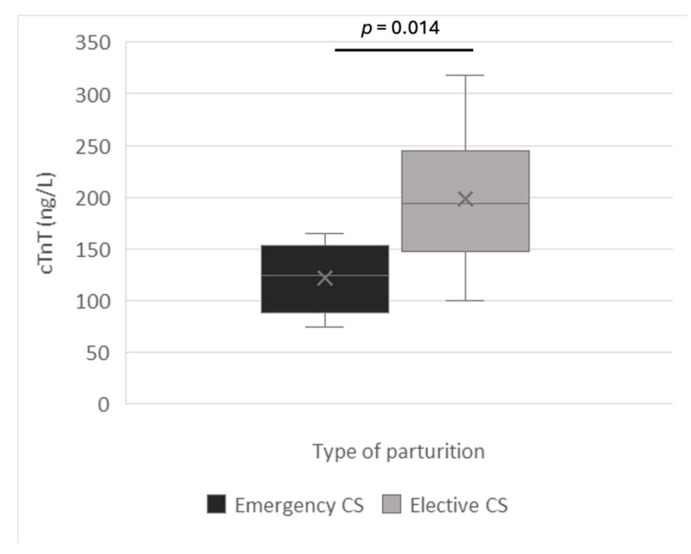
cTnT concentrations in relation to the type of parturition. CS means caesarean section.

**Figure 4 vetsci-12-00952-f004:**
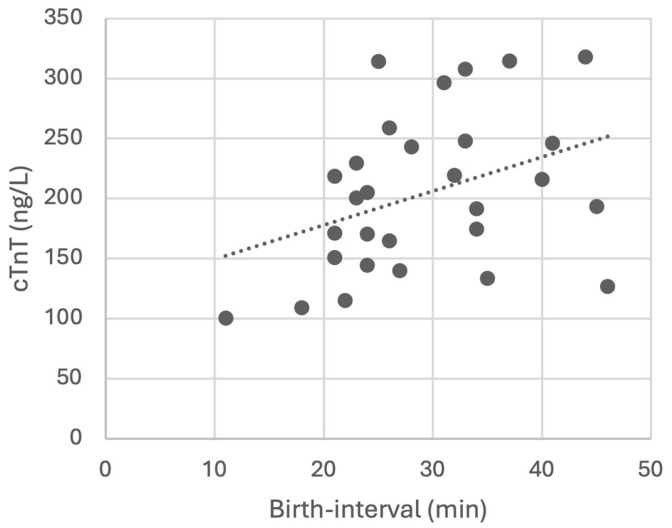
cTnT concentrations in relation to birth interval. Dots represent individual data points; the dashed line represents the trend.

**Figure 5 vetsci-12-00952-f005:**
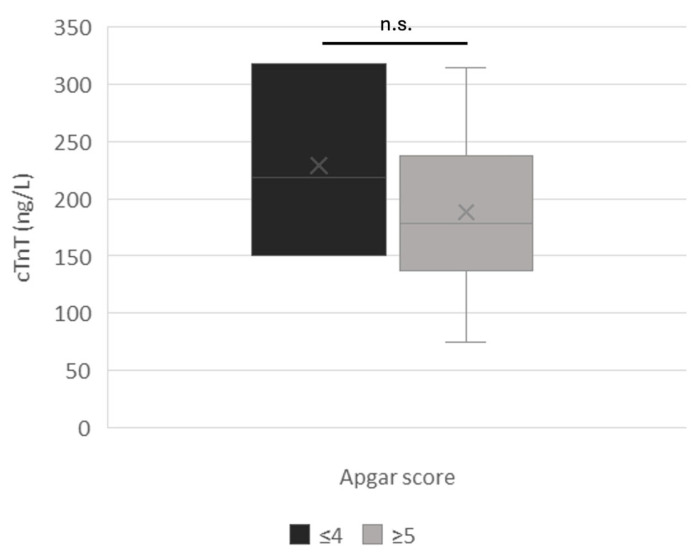
cTnT amniotic concentrations in relation to Apgar score.

**Table 1 vetsci-12-00952-t001:** Bitches enrolled in this study.

ID.	Breed	Age (Year)	BW (kg)	Litter Size	Type of Parturition
1	Bernese Mountain dog	5.6	45.1	2	Elective CS
2	Bernese Mountain dog	3.5	62.7	16	Elective CS
3	Boxer	5	29	2	Elective CS
4	Chihuahua	4.6	3.3	1	Emergency CS
5	Chihuahua	4.4	3	2	Emergency CS
6	Chihuahua	3.3	3.1	2	Natural
7	Chihuahua	7	3.8	1	Emergency CS
8	Chihuahua	5.6	2.4	2	Elective CS
9	English Bull dog	5.3	27.2	2	Elective CS
10	Entlebucher Mountain dog	6.5	26.4	2	Elective CS
11	French Bull dog	4.3	9	3	Elective CS
12	Poodle	4.6	4	1	Elective CS
13	Staffordshire Bull terrier	8.5	16.2	1	Emergency CS
14	Welsh Corgi	6	15.8	3	Elective CS
	Mean ± sd	5.3 ± 1.4	17.9 ± 18.4	3 ± 3.8	

BW means body weight; CS means caesarean section.

## Data Availability

The original contributions presented in this study are included in the article. Further inquiries can be directed to the corresponding authors.

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
