# Peer review of "First Report on Cardiac Troponin T Detection in Canine Amniotic Fluid†"

_vetsci, 2025, doi:10.3390/vetsci12100952_

Round 1

Reviewer 1 Report

Comments and Suggestions for Authors

This manuscript presents a novel pilot study investigating the presence of cardiac troponin T (cTnT) in canine amniotic fluid (AF) collected at birth and its potential associations with maternal and neonatal parameters. The paper is clearly written, with a logical flow from introduction to conclusions. Methods wer carefully described, and ethical approval was obtained. The statistical analysis highlights significant correlations between cTnT and maternal morphotype, age, weight, type of parturition, and birth interval. These findings support the hypothesis that cTnT may reflect peripartum stress or hypoxia. Limitations are related to the sample size: only 14 bitches and 40 puppies were included, which limits the generalizability of the results and the robustness of statistical associations. I suggest expanding the study with a larger number of subjects included (bitches and newborns) and diversifying the sample, including different breeds.

Author Response

This manuscript presents a novel pilot study investigating the presence of cardiac troponin T (cTnT) in canine amniotic fluid (AF) collected at birth and its potential associations with maternal and neonatal parameters. The paper is clearly written, with a logical flow from introduction to conclusions. Methods wer carefully described, and ethical approval was obtained. The statistical analysis highlights significant correlations between cTnT and maternal morphotype, age, weight, type of parturition, and birth interval. These findings support the hypothesis that cTnT may reflect peripartum stress or hypoxia. Limitations are related to the sample size: only 14 bitches and 40 puppies were included, which limits the generalizability of the results and the robustness of statistical associations. I suggest expanding the study with a larger number of subjects included (bitches and newborns) and diversifying the sample, including different breeds.

R/ Thank to the Reviewer for the constructive and encouraging feedback on our research. Regarding the suggestion to include additional cases, we would like to clarify that the primary aim of this pilot study was to explore the presence of troponin in canine amniotic fluid, a finding that, to the best of our knowledge, has not been previously reported in this species. While we fully acknowledge that several clinical aspects still require further investigation, the measurement of cardiac troponin T in amniotic fluid immediately after birth may represent a promising diagnostic tool for the early identification of critically ill newborn puppies. Limitations on the sample size of our caseload have been further highlighted in the manuscript. We hope this helps to clarify the focus and scope of our study. As requested, the English language has been thoroughly revised to improve clarity and readability.

Reviewer 2 Report

Comments and Suggestions for Authors

The study by Eliza Giussani and colleagues addresses the novel issue of detecting cardiac troponin T in the amniotic fluid of female dogs. To the best of current knowledge, this is the first report in the scientific literature describing such a diagnostic approach in canines. The topic is both original and highly relevant, particularly in the context of early diagnosis of neonatal disorders in puppies. Neonatal mortality remains alarmingly high, with a considerable proportion of puppies dying within the first days after birth, and the underlying causes often remain unidentified. In this regard, the detection of cardiac troponin T in amniotic fluid immediately after birth could contribute to earlier recognition of critical conditions and thereby facilitate the implementation of preventive measures, potentially reducing postpartum mortality in puppies. This applies regardless of whether delivery is natural or performed via elective or emergency caesarean section.

The study, however, was conducted on a relatively small cohort of female dogs of varying breeds, weights, and ages. This limitation reduces the statistical power of the findings and should be considered when interpreting the results. Nonetheless, it does not diminish the potential importance of this pioneering work, which may ultimately provide a valuable tool for early diagnosis of hypoxia, anemia, and other perinatal disorders in newborn puppies. Importantly, the authors themselves acknowledge the limitations of their study, as discussed in the article’s discussion and conclusions.

The results indicate that factors such as maternal weight, age, duration of labor, and mode of delivery may significantly influence cardiac troponin T levels. Therefore, expanding the sample size in future studies could refine or potentially alter these preliminary findings. The research methodology is clearly described, and the presentation of the results (one table and four figures) is appropriate and transparent. Furthermore, the cited references are current, relevant, and aligned with the scope of the presented research.

Overall, despite certain methodological limitations, the study offers promising insights that may contribute to a better understanding of neonatal health in dogs and potentially reduce the high mortality rates observed in puppies.

Author Response

The study by Eliza Giussani and colleagues addresses the novel issue of detecting cardiac troponin T in the amniotic fluid of female dogs. To the best of current knowledge, this is the first report in the scientific literature describing such a diagnostic approach in canines. The topic is both original and highly relevant, particularly in the context of early diagnosis of neonatal disorders in puppies. Neonatal mortality remains alarmingly high, with a considerable proportion of puppies dying within the first days after birth, and the underlying causes often remain unidentified. In this regard, the detection of cardiac troponin T in amniotic fluid immediately after birth could contribute to earlier recognition of critical conditions and thereby facilitate the implementation of preventive measures, potentially reducing postpartum mortality in puppies. This applies regardless of whether delivery is natural or performed via elective or emergency caesarean section. The study, however, was conducted on a relatively small cohort of female dogs of varying breeds, weights, and ages. This limitation reduces the statistical power of the findings and should be considered when interpreting the results. Nonetheless, it does not diminish the potential importance of this pioneering work, which may ultimately provide a valuable tool for early diagnosis of hypoxia, anemia, and other perinatal disorders in newborn puppies. Importantly, the authors themselves acknowledge the limitations of their study, as discussed in the article’s discussion and conclusions. The results indicate that factors such as maternal weight, age, duration of labor, and mode of delivery may significantly influence cardiac troponin T levels. Therefore, expanding the sample size in future studies could refine or potentially alter these preliminary findings. The research methodology is clearly described, and the presentation of the results (one table and four figures) is appropriate and transparent. Furthermore, the cited references are current, relevant, and aligned with the scope of the presented research. Overall, despite certain methodological limitations, the study offers promising insights that may contribute to a better understanding of neonatal health in dogs and potentially reduce the high mortality rates observed in puppies.

R/ We sincerely thank the Reviewer for taking the time to evaluate our manuscript and for the positive feedback. We have included their relevant observations in the revised text.

Reviewer 3 Report

Comments and Suggestions for Authors

This manuscript presents the first investigation of cardiac troponin T (cTnT) concentrations in canine amniotic fluid, aiming to explore its potential as a biomarker of neonatal vitality. The topic is original and clinically relevant, and the manuscript is generally well written, clearly structured, and supported by appropriate methodology and statistical analyses. Well-described methodology, including ethical approval and standardized sampling. Data clearly presented with tables and figures, and an adequate discussion of possible confounding factors. However, in my opinion, some points could be improved. The number of bitches is small and heavily unbalanced in terms of breed, morphotype, and litter size. This limitation should be explicitly highlighted in the Discussion; maternal weight, age, and morphotype are all interrelated and may confound each other’s effects. A brief comment on this and the absence of multivariate analysis would be helpful. The study would benefit from a clearer statement about the possible clinical utility of amniotic cTnT (e.g., potential cut-off values to be explored in future studies, diagnostic sensitivity/specificity expected).

Author Response

This manuscript presents the first investigation of cardiac troponin T (cTnT) concentrations in canine amniotic fluid, aiming to explore its potential as a biomarker of neonatal vitality. The topic is original and clinically relevant, and the manuscript is generally well written, clearly structured, and supported by appropriate methodology and statistical analyses. Well-described methodology, including ethical approval and standardized sampling. Data clearly presented with tables and figures, and an adequate discussion of possible confounding factors. However, in my opinion, some points could be improved. The number of bitches is small and heavily unbalanced in terms of breed, morphotype, and litter size. This limitation should be explicitly highlighted in the Discussion; maternal weight, age, and morphotype are all interrelated and may confound each other’s effects. A brief comment on this and the absence of multivariate analysis would be helpful. The study would benefit from a clearer statement about the possible clinical utility of amniotic cTnT (e.g., potential cut-off values to be explored in future studies, diagnostic sensitivity/specificity expected).

R/ Thanks to the Reviewer for raising this important point. We agree that maternal weight, age, morphotype, and litter size are interrelated variables that may confound each other’s effects on amniotic cTnT concentrations. In this pilot study, however, the relatively small and unbalanced sample size did not allow us to reliably perform a multivariate analysis without the risk of overfitting and spurious associations. These considerations and limitations of our study have been included and highlighted in the revised manuscript to better clarify the scope and context.

Round 2

Reviewer 1 Report

Comments and Suggestions for Authors

Thank you for the revision.